# OpenReview forum: "RewardBench: Evaluating Reward Models for Language Modeling"
_NeurIPS.cc/2024/Datasets_and_Benchmarks_Track — Submitted to NeurIPS 2024 Track Datasets and Benchmarks_

### Official Review · Reviewer_RUcU · 2024-07-10

**Rating:** 9
**Confidence:** 4
**Clarity:** The paper is very well written.

**Review:**

Overall, I believe this work is a huge service to the community.
Pros: The work is highly extensive, setting the community up for iteration, allowing both practitioners and researchers to conduct many more future works in this space.
Cons: It would have been wonderful to have more documentation on the types of datasets each reward model is trained on.

These are detailed in the below fields.

**Strengths:**

In particular, for anyone who would like to train a reward model and compare it to existing ones, this work makes it as easy as running a few lines of code and iterate. I feel very strongly positive about this paper, and I believe it paves the way for a lot more work. In particular,
- One can study the effects architecture/ other ablations in training reward models
- One can study the effects of data composition for reward models (points for improvement)
- One can determine what off-the-shelf model to use as reward across publicly available ones.

In general,
- the work is highly extensive evaluating tens of reward models on thousands of prompts taking the load of such evaluations from community and provides these artifacts
- thoughtfully develops a suite of datasets to test reward models
- allowing many future works in this space.

**Additional Feedback:**

Thank you for your service to the community.

**Correctness:**

I do believe the claims are correct and the datasets are curated in a sound way.

**Documentation:**

There is a lot of documentation on huggingface and the official repository of the benchmark.

**Ethics:**

I do not.

**Limitations:**

I do not see any negative impact beyond what is discussed.

**Opportunities For Improvement:**

I think the most important point to improve on is the documentation for the datasets the existing reward models are trained on. This way, we would be able to evaluate the impact of dataset composition and size on the performance of the reward models in different subsets of the benchmark. e.g., how does having multiple languages in the preference datasets to train RMs impact performance? How much data do we need from different domains to perform well within domain? How does domain transfer work, in that does having data from specific domains help the performance in other domains?

In general, I do not expect this work to answer all these questions in addition to all the extensive work being done. However, if these information was in the package, or if there was a way to populate these pieces of documentation, I think it would be a huge service to the community to be able to study these questions easily.

**Relation To Prior Work:**

The relation to prior work is well discussed.

**Summary And Contributions:**

This work proposes a large scale and extensive benchmark to evaluate reward models. In particular, reward models are used to improve LLMs, and sometimes used as process-reward-models even to provide step-by-step supervision. Even though models are optimized for these rewards, the public research community does not have a good understanding of what constitutes a good reward model, or there has not been an extensive and publicly available resource to set ourselves up for iteration. RewardBench is an open-source benchmark that evaluates many types of rewards models across a large suite, which is an invaluable resource for the community.

---

> ### Author Rebuttal · Authors · 2024-08-14
>
> Thank you for your encouraging words and helpful feedback. We are glad to hear you found the paper well written and the tooling to be a large opportunity for our community.
>
> We’ve added a table of the training datasets used for the top models to the paper and are investigating ways to integrate it seamlessly into the leaderboard. The added table, converted into markdown for the rebuttal (we had to remove rows to fit within the character limit).
>
> We agree that data mixing and its impacts are of the utmost importance to the downstream evolution of reward model training and evaluation. We highlight the two data points in the paper:
> 1. How many models are trained on the "Zephyr" recipe, using the UltraFeedback dataset, especially with DPO.
> 2. DPO models tend not to be trained on the datasets in "Prior Sets," which can make the gap from DPO to classifiers seem larger.
>
> Now that the leaderboard is more populated, running more advanced analyses in this area would be a great area of followup work.
>
> If there are any further questions you have after seeing the other reviews, please let us know.
>
>
> | **Reward Model** | **Preference Datasets Used** |
> |------------------|------------------------------|
> | [RLHFlow/ArmoRM-Llama3-8B-v0.1](https://huggingface.co/RLHFlow/ArmoRM-Llama3-8B-v0.1) | [HelpSteer](https://huggingface.co/datasets/nvidia/HelpSteer), [UltraFeedback](https://huggingface.co/datasets/openbmb/UltraFeedback), BeaverTails, Argilla-Capybara, Argilla-Math-Preferences, CodeUltraFeedback, Argilla-OpenOrca |
> | [RLHFlow/pair-preference-model-LLaMA3-8B](https://huggingface.co/RLHFlow/pair-preference-model-LLaMA3-8B) | Filtered [HH-RLHF](https://huggingface.co/datasets/Anthropic/hh-rlhf), [SHP](https://huggingface.co/datasets/stanfordnlp/SHP), [HelpSteer](https://huggingface.co/datasets/nvidia/HelpSteer), SafeRLHF-30k, [UltraFeedback](https://huggingface.co/datasets/openbmb/UltraFeedback), UltraInteract, CodeUltraFeedback, Argilla-Math, OpenOrca, Capybara |
> | [sfairXC/FsfairX-LLaMA3-RM-v0.1](https://huggingface.co/sfairXC/FsfairX-LLaMA3-RM-v0.1) | Filtered [HH-RLHF](https://huggingface.co/datasets/Anthropic/hh-rlhf), [SHP](https://huggingface.co/datasets/stanfordnlp/SHP), [HelpSteer](https://huggingface.co/datasets/nvidia/HelpSteer), SafeRLHF-30k, [UltraFeedback](https://huggingface.co/datasets/openbmb/UltraFeedback), UltraInteract, CodeUltraFeedback, Argilla-Math, OpenOrca, Capybara |
> | [openbmb/Eurus-RM-7b](https://huggingface.co/openbmb/Eurus-RM-7b) | UltraInteract, [UltraFeedback](https://huggingface.co/datasets/openbmb/UltraFeedback), UltrSafety |
> | [Nexusflow/Starling-RM-34B](https://huggingface.co/Nexusflow/Starling-RM-34B) | Nectar |
> | [weqweasdas/RM-Mistral-7B](https://huggingface.co/weqweasdas/RM-Mistral-7B) | [HH-RLHF](https://huggingface.co/datasets/Anthropic/hh-rlhf), Capybara, Orca, [SHP](https://huggingface.co/datasets/stanfordnlp/SHP), [UltraFeedback](https://huggingface.co/datasets/openbmb/UltraFeedback), HelpSetter, PKU-SafeRLHF, PKU-SafeRLHF-30k |
> | [hendrydong/Mistral-RM-for-RAFT-GSHF-v0](https://huggingface.co/hendrydong/Mistral-RM-for-RAFT-GSHF-v0) | **Undisclosed** |
> | [stabilityai/stablelm-2-12b-chat](https://huggingface.co/stabilityai/stablelm-2-12b-chat) | [HH-RLHF](https://huggingface.co/datasets/Anthropic/hh-rlhf), argilla/dpo-mix-7k, and other **Undisclosed** |
> | [Ray2333/reward-model-Mistral-7B-instruct...](https://huggingface.co/Ray2333/reward-model-Mistral-7B-instruct...) | [Summarize](https://huggingface.co/datasets/openai/summarize_from_feedback), [WebGPT](https://huggingface.co/datasets/openai/webgpt_comparisons), Dahoas/instruct-synthetic-prompt-responses, [HH-RLHF](https://huggingface.co/datasets/Anthropic/hh-rlhf), ChatBotArena Conversations, [UltraFeedback](https://huggingface.co/datasets/openbmb/UltraFeedback), Nectar |
> | [allenai/tulu-2-dpo-70b](https://huggingface.co/allenai/tulu-2-dpo-70b) | UltraFeedback |
> | [meta-llama/Meta-Llama-3-70B-Instruct](https://huggingface.co/meta-llama/Meta-Llama-3-70B-Instruct) | **Undisclosed** |
> | [prometheus-eval/prometheus-8x7b-v2.0](https://huggingface.co/prometheus-eval/prometheus-8x7b-v2.0) | [Preference Collection](https://huggingface.co/datasets/prometheus-eval/Preference-Collection) (relabeled mix) |
> | [NousResearch/Nous-Hermes-2-Mistral-7B-DPO](https://huggingface.co/NousResearch/Nous-Hermes-2-Mistral-7B-DPO) | **Undisclosed** |
> | [mistralai/Mixtral-8x7B-Instruct-v0.1](https://huggingface.co/mistralai/Mixtral-8x7B-Instruct-v0.1) | **Undisclosed** |
> | [upstage/SOLAR-10.7B-Instruct-v1.0](https://huggingface.co/upstage/SOLAR-10.7B-Instruct-v1.0) | OpenOrca, Intel-Orca, UltraFeedback |
> | [HuggingFaceH4/zephyr-7b-alpha](https://huggingface.co/HuggingFaceH4/zephyr-7b-alpha) | UltraFeedback |
> | [allenai/tulu-2-dpo-13b](https://huggingface.co/allenai/tulu-2-dpo-13b) | UltraFeedback |
> | [0-hero/Matter-0.1-7B-boost-DPO-preview](https://huggingface.co/0-hero/Matter-0.1-7B-boost-DPO-preview) | **Undisclosed** |
> | [prometheus-eval/prometheus-7b-v2.0](https://huggingface.co/prometheus-eval/prometheus-7b-v2.0) | [Preference Collection](https://huggingface.co/datasets/prometheus-eval/Preference-Collection) (relabeled mix) |
> | [HuggingFaceH4/starchat2-15b-v0.1](https://huggingface.co/HuggingFaceH4/starchat2-15b-v0.1) | [UltraFeedback](https://huggingface.co/datasets/openbmb/UltraFeedback), Orca |
> | [HuggingFaceH4/zephyr-7b-beta](https://huggingface.co/HuggingFaceH4/zephyr-7b-beta) | UltraFeedback |
> | [allenai/tulu-2-dpo-7b](https://huggingface.co/allenai/tulu-2-dpo-7b) | UltraFeedback |

---

> > ### Comment · Reviewer_RUcU · 2024-08-19
> > **Response**
> >
> > Thank you for the response, I appreciate the details on the preference datasets. I'll maintain my score.

---

### Official Review · Reviewer_T6o3 · 2024-07-25

**Rating:** 4
**Confidence:** 5
**Correctness:** Yes.
**Clarity:** Yes.

**Review:**

From an engineering perspective, this paper significantly contributes to the current evaluation of Reward Models, providing a broad assessment of existing models.

Weaknesses:

1. This work relies on the integration of existing preference datasets. From the viewpoint of scientific contribution, such efforts might be more appropriately attributed to the original papers, such as Alpaca Eval or MT Bench. As a reviewer, the aggregative nature of this work lacks the innovation required to meet the standards of NeurIPS DB.

2. A significant concern from the reviewer’s perspective is that RLHFlow/ArmoRM-Llama3-8B-v0.1 is only an 8B model, yet it achieves a performance score of 90 on this leaderboard. This raises questions about the leaderboard's level of challenge.

3. The reviewer acknowledges that unique analyses of current Reward Models based on different datasets can be insightful. However, the analyses presented from Table 1 to Table 5 are weak, merely reiterating a common phenomenon without exploring the differences across various training methods and model sizes.

4. From a licensing perspective, it is curious whether the authors obtained permission to directly aggregate data from other research papers without significant refinement or modifications that demonstrate substantial additional effort.

From the reviewers' perspective, although this paper has garnered some attention within the community, it does not meet the reviewers' bar for scientific rigor and innovation required for acceptance. The approach of directly aggregating existing work without further refinement or substantive improvements may contribute from an engineering standpoint, but it falls short in terms of scientific contribution.

**Strengths:**

See above.

**Additional Feedback:**

No.

**Documentation:**

Yes.

**Ethics:**

I am concerned whether this paper has scientific contributions and whether it encroaches on the scientific contributions of existing work.

**Limitations:**

I am concerned whether this paper has scientific contributions and whether it encroaches on the scientific contributions of existing work.

**Opportunities For Improvement:**

See above.

**Relation To Prior Work:**

No.

**Summary And Contributions:**

This paper present Reward Bench, a benchmark dataset and code-base for evaluation,  including easoning, and safety, to benchmark how reward models perform on challenging, structured and out-of-distribution queries.

---

> ### Author Rebuttal · Authors · 2024-08-14
>
> We’ve addressed each of your comments below. We begin with the fourth point and the related ethics flag because it’s extremely important to clarify.
>
> ### 4. Regarding the licenses, permissions to use datasets, and ethics review
>
> We took very careful consideration on the provenance and permission to use the datasets in this benchmark. The license details are clearly include in the dataset card (and we have also added this to the paper for the final version). https://huggingface.co/datasets/allenai/reward-bench#license-information
>
> The license we use is ODC-By, which indicates that the user is responsible for complying with the licenses of the subsequent parts of the dataset. Let us know if you have any other concerns. In fact, we have had discussions with most of the datasets’ authors, and have never heard anything but positive feedback. Aggregation of existing datasets, with attribution, is very much a norm in this research area.
>
> ### 1. Regarding novelty and dataset curation
>
> Please see the extensive comments in the general comment to reviewers. Building on and modifying existing datasets is a common practice crucial to progress in ML research. We also want to point out that the idea of using pairwise scores as an accuracy for reward models is entirely new, so identifying how to do this is a contribution on its own.
>
> We would agree with the reviewer that just mixing datasets is alone not enough for a dataset paper, but ours is a substantial effort that delivers new benchmarking infrastructure AND improved datasets through manual curation.
>
> ### 2. Regarding saturation
>
> We also addressed this at length in the general comment. Would it be better if we normalized the evaluation scores from 0 to 1, where 50% (random) is 0 and 100% is 1, with the potential of negative scores? This might make the lack of saturation, especially on Chat Hard, clearer.
>
> ### 3. Regarding training insights and trends on reward models
>
> We do document the differences between many types of models and sizes throughout the paper (and in the leaderboard). We acknowledge that there is more work to be done on how to best train these models, but as a starting point believe we have made substantial progress in the understanding of reward model usages in language models.

---

> > ### Comment · Reviewer_T6o3 · 2024-08-24
> > **reply to the author**
> >
> > see above.

---

> > > ### Comment · Reviewer_T6o3 · 2024-08-24
> > > **reply to the author**
> > >
> > > The author’s attitude does not give me the impression that their behavior is appropriate for a rebuttal. It seems they take the acceptance of this work for granted and have not focused on addressing the reviewers' concerns. I have decided to increase my confidence level and stand by my review.

---

### Official Review · Reviewer_7Cs6 · 2024-07-25

**Rating:** 5
**Confidence:** 4
**Correctness:** yes.
**Clarity:** yes.

**Review:**

1. The paper introduces a comprehensive evaluation framework for assessing various reward model architectures, accompanied by visualization, training, and other analytical tools.
2. The paper provides a detailed comparison of the performance differences between DPO models and classifier-based reward models across various datasets. Although DPO models are more common due to their simplicity, they fail to generalize well to popular preference data test sets and exhibit higher performance variance.

**Strengths:**

Weaknesses:

1. The authors evaluate safety RMs. What is the distribution of these data across different safety categories? For example, references [1][2] provide a detailed categorization of safety. How evenly are the 740 Safety data points distributed across different categories, and do they cover a sufficient range of scenarios?

[1]. SafetyBench: Evaluating the Safety of Large Language Models: https://arxiv.org/abs/2309.07045

[2]. BeaverTails: https://huggingface.co/datasets/PKU-Alignment/PKU-SafeRLHF

2. The authors utilized the Anthropic HHH and HH-RLHF datasets, which were open-sourced in 2021 and early 2022. These datasets have lower quality prompts and preference pairs. For instance, using hh-rlhf for DPO or PPO results in performance degradation. Did the authors apply any filtering strategies, and have they considered using higher-quality safety and helpfulness preference datasets?

3. In the Related Work section, the authors discuss AlpacaFarm, MT-Bench, and others. Besides effectively aggregating datasets, does this paper offer any unique innovations in evaluation methodology?

4. A concern of mine is that while the authors have aggregated existing datasets, they have not organized the subcategories of these datasets in a fine-grained manner, such as safety classifications. As mentioned in the first point, the evaluation appears to be broad rather than fine-grained.

Another concern is the scientific contribution of this paper. While I find it somewhat lacking, considering this is a DB track, I look forward to the opinions of other reviewers, especially regarding the academic contribution of this paper.

**Additional Feedback:**

see above.

**Documentation:**

yes.

**Ethics:**

no.

**Limitations:**

yes.

**Opportunities For Improvement:**

see above.

**Relation To Prior Work:**

yes.

**Summary And Contributions:**

The authors propose Reward Bench, a benchmark that aggregates multiple perspectives of Reward Model evaluations, including Chat, Safety, Reasoning, and more.

---

> ### Author Rebuttal · Authors · 2024-08-14
>
> Thank you for your comments on how we can improve RewardBench and on how it is a comprehensive framework for starting to evaluate reward models (and DPO trained models). We’ve included detailed responses to your questions below.
>
> ### 1. On safety evaluation.
>
> Specifically regarding BeaverTails (the safe RLHF dataset), we evaluated agreement on this dataset, but found agreement to be low and results closer to random, so we decided not to include it on the leaderboard. We could include the scores on the prior-tests tab if you think it is important.
> The code where this is done is here: https://github.com/allenai/reward-bench/blob/bc72fb2a573fc31c614eef3405d354b398977b02/rewardbench/utils.py#L316
>
> Regarding Safety fine-grained categories, we decided that is out of scope due to the goal of getting an initial reward modeling evaluation infrastructure done for the community with the hopes of more specialized reward modeling evaluation being built in the future. In the hope of building community adoption around reward models, which was severely lacking when we submitted this paper, we thought a single score rather than fine-grained was preferred.
>
> ### 2. On HH-RLHF datasets.
>
> In previous work, agreement on these datasets was used as a benchmark for reward models without filtering. We wanted the prior sets to be a bridge between previous work and a new generation of reward model benchmarks.
>
> Second, the point you make is why we reduced the weight of prior sets for the paper to 0.5 weight. In the current leaderboard, the Prior Sets are not included by default due to their slightly outdated nature.
>
> ### 3. On data curation practices.
>
> Please see our general comment above and the Appendix, and let us know if you have any further questions.
>
> ### 4. On fine-grained evaluation.
>
> By zooming into the subsets that make up the sections, there are 27 fine-grained categories, but more work is needed because they currently overlap in topics. We think evaluation is a very challenging task, especially in a fast moving space, so it is better to try and align evaluations with research goals rather than trying to solve every component with one tool. We are hoping to be the first general evaluation that people look at, but finer grained evaluations built soon will provide more clarity for specific domains.
>
> ### 5. On scientific contribution.
>
> We tried to include many experiments, such as the DPO reference free computation strategies and generative language models, which paint the space of reward models in a new light. There are further results in the Appendix. Let us know if there are any specific questions you may have. We think that the paper is a good fit for Datasets and Benchmarks, rather than the main track where building on the experiments we did would be more important.
>
> The direct evaluation of RMs, other than extrinsic downstream evaluations, is an underexplored research area and what exactly is encoded in (preference) RMs and what their pitfalls are often remains opaque. A large part of our scientific contribution therefore is in simply reckoning with the question of “how should one evaluate a reward model?”

---

> > ### Comment · Reviewer_7Cs6 · 2024-08-21
> > **reply to the author**
> >
> > Thank you very much for your response. I agree with the concerns raised by Reviewer T6o3 and choose to lower my score. I think this work should have more unique innovation.
> >
> > I also read the review results of Reviewer RUcU and agree with his perspective on this work. However, he gave this work a score of 9, which in my opinion is far from my evaluation.

---

> > > ### Author Response · Authors · 2024-08-23
> > > **Response to reviewer 7Cs6**
> > >
> > > Please see our above comment on the unique innovations of this work or let us know if you have more specific areas of feedback. We disagree that the data work is not innovative and offer many other areas of contribution.

---

### Author Rebuttal · Authors · 2024-08-14

We appreciate the time and effort the reviewers put into understanding and critiquing our paper. In their reviews there are a few common themes that we will address here in the general response, and then we will respond to each reviewer individually in more depth. The two points we want to clarify in the general rebuttal are 1) how we curated the data beyond naively aggregating existing datasets and 2) how to interpret the scores and the potential saturation of the leaderboard already.

## On data curation practices, efforts, and contributions

To start, carefully aggregating datasets in novel manners is often seen as a scientific contribution in many areas of machine learning. Examples include AlpacaEval, Glue, SuperGlue, and Paloma. We went beyond this in RewardBench, with extensive details in the Appendix.

We did a substantial amount of work to prepare the datasets and carefully build on existing academic evaluation datasets. In order to build a benchmark, we only considered datasets that have a clear test set to avoid potential contamination in existing training data. From there, for every subset we considered (i.e. the minor breakdown of sections Chat, Chat Hard, etc.) we performed some amount of manual labor. These details are included in the appendix as `Data processing notes` – the final section. Essentially, we used manual labor to either verify existing heuristics make sense and in some cases to manually write out completions to models.

For example, for the AlpacaEval and MT Bench subsets, we checked every sample’s conformation to a set of data label principles (Appendix J.1 in update version) to make sure filtering was not biasing the dataset nor leaving in erroneous data. Other examples include the safety subsets refusals-dangerous and refusals-offensive, we needed to experiment with multiple models to find one that could generate completions that reliably did not refuse the requests, and then we manually verified each response. For some of the XSTest subsets, we added manually written examples and examples from multiple open models when the heuristics for matching pairs did not yield any samples for our evaluation.

Naive aggregation resulted in saturated scores similar to what is seen in Prior Sets, where the agreement on some subsets caps out at 75 to 80 percent. This manual labor was crucial to making a useful benchmark, but was largely documented in the appendix to allow space for more results.

## On score saturation and long term relevance

This project has had a rapidly evolving lifecycle, but at the time of review there is still a lot of usefulness left in the benchmark. To give some context, when this project was first launched, just before submission, the top open reward models were scoring only at about 65 percent out of 100. The quick uptake of the benchmark and improvement of scores largely seems to validate our results.

Even as new frontier models at top labs come out, such as GPT-4o or Gemini 1.5 versions, they regularly creep higher on the benchmark. The correlation between many types of models based on downstream usefulness has exceeded our expectations and we see continued potential with it. Regardless, we acknowledge that a large part of this contribution is in building tooling and workflows that we hope will be used for future datasets that more specifically challenge models.

At quick glance, with the top models scoring around 89% accuracy, it seems like the benchmark may already be completed, but the relevant score is only from 50 to 100, rather than from 0.

Furthermore, we acknowledge that the sections Chat, Safety, and Reasoning have been solved more easily by models. In terms of open and smaller models (which are of course important to the academic research community), there is still substantial progress to be made on the Chat Hard category.

Getting one score that was calibrated with strong models is the most important part of a benchmark, enabling clear communication and iteration (related reading https://www.safe.ai/blog/devising-ml-metrics). Until there are many models that score well on all subsets, including Chat Hard where even some top models struggle (such as Llama 3.1 70b getting in the 65% range), the benchmark will continue to be of use to the community.

## The novelty of building the first benchmark

In order to kickstart progress in the academic community, building a high-quality general benchmark is needed. To the best of our knowledge, we were the first to propose a benchmark for evaluating preference reward models. The process for getting here and how to use these datasets was entirely unestablished in the community. Setting norms for how to benchmark reward models at all is an important contribution of this paper. After this first benchmark, we agree that future evaluation tools should be considered with more precision and in the context of methods we have established. To our knowledge, we are the first group to directly compare DPO models to standard reward models at classification tasks.

---

### Author Response · Authors · 2024-08-23
**Response to reviewers on novelty & innovation**

We would like to strongly argue for the novelty and unique innovation of our work. To our knowledge, this is the first project to evaluate and compare reward models directly. Throughout the code and report there are many unique details that we had to create in order to make this possible. It would help if these reviewers could provide examples on what more could be done to strengthen our claims (in the context of length limitations).

RewardBench has already been embraced by the academic and industrial communities building both open and closed frontier models. Google, Nvidia, Apple, Perplexity, Meta, and others have already submitted models or built on our tools. There are numerous papers expanding reward model training that heavily utilize and analyze our submission further (such as https://arxiv.org/abs/2405.07863, https://arxiv.org/abs/2407.06551, https://arxiv.org/abs/2408.02666, https://arxiv.org/abs/2408.11791, and others ). In fact, RewardBench has been downloaded greater than 200,000 times in the last month. It is clear that there is community demand across the entire field for the innovation we put substantial effort and creativity into.

We feel in the context of a benchmarks track, this is in line with expected levels of scientific novelty. Similar efforts to combine and improve existing datasets have been established as innovative, as we mention above with Glue, AlpacaEval, and others.

---

> ### Comment · Reviewer_T6o3 · 2024-08-24
> **reply to the author**
>
> I believe the author's discussion does not focus on the work at hand, and discussing other works is not appropriate in this context.
>
> Firstly, Glue has specific, unique advantages. Without demonstrating the innovation and impact of this work, **I do not think RewardBench can be objectively compared to Glue.**
>
> Secondly, regarding AlpacaEval, could the authors clarify which conference or journal has accepted this dataset?
>
> Moreover, the emphasis on the download count of RewardBench is not necessarily related to the innovation and acceptability of the current submission.
>
> I have consistently stressed that the design of RewardBench lacks innovation and may hinder the community's creativity. If combining existing datasets like ax+by+cz can be accepted by the community, why would anyone bother to innovate new datasets? The community might start focusing on various combinations of existing datasets instead of fostering true innovation.
>
> While I acknowledge the engineering effort in this work, I still do not believe this dataset is worthy of acceptance unless it can demonstrate uniqueness.

---

### Decision · Program_Chairs · 2024-09-26

**Decision:**

Reject

**Comment:**

This paper introduces RewardBench, a benchmark dataset and codebase for evaluating reward models (RMs) used in reinforcement learning from human feedback (RLHF). The authors argue that while RLHF has been instrumental in aligning large language models (LLMs) with human preferences, there has been a lack of focus on evaluating the RMs themselves. They address this gap by providing a comprehensive suite of evaluation tools, including a dataset of prompt-win-lose trios and a leaderboard for comparing different RMs. The dataset covers a range of tasks, including chat, reasoning, and safety, and the leaderboard evaluates RMs trained with various methods, such as direct MLE training and DPO. The authors also present findings on the limitations of current RMs, such as their propensity for refusals, reasoning limitations, and instruction-following shortcomings.

However, there are some concerns that warrant attention. The reliance on "semi-automatic" methods for data collection, without a clear explanation of these methods or their potential limitations, raises questions about the dataset's quality and potential biases. It would be beneficial for the authors to elaborate on these methods and discuss any steps taken to mitigate potential issues. Additionally, the potential for spurious correlations in reasoning tasks, particularly in cases where only a few tokens differ between the chosen and rejected samples, could impact the validity of the evaluation results. The authors could address this by providing more diverse and challenging reasoning examples or by exploring alternative evaluation metrics that are less susceptible to such correlations. Reviewers' concerns about data contamination, where models might be inadvertently trained on the evaluation data, should be addressed.

Despite these limitations, this paper makes a significant contribution by introducing the first benchmark dataset and codebase for evaluating reward models. The open-source release of the code and data will undoubtedly facilitate further research and development in this important area. The insights provided on the limitations of current RMs and the challenges in evaluating them are valuable for guiding future work. I encourage the authors to consider the feedback provided and continue to refine and expand upon this work, potentially addressing the concerns raised and further exploring the connection between benchmark results and downstream RLHF performance. While the paper addresses an important gap in the field, the limitations and open questions suggest that further development and refinement are necessary before it can be considered a strong contribution to the conference.

Pros:
- Addresses a crucial gap: The paper tackles the under-explored area of RM evaluation, which is essential for understanding and improving RLHF methods.
- Comprehensive evaluation tools: The benchmark dataset and leaderboard provide a valuable resource for researchers and practitioners to assess and compare different RMs.
- Diverse tasks and models: The dataset covers a wide range of tasks, and the leaderboard includes RMs trained with different methods and on different datasets, providing a broad overview of the current state of the art.
- Insightful findings: The paper presents several interesting findings on the limitations of current RMs, which can guide future research in this area.
- Open-source code and data: The release of the code and data facilitates reproducibility and further research in this field.

Cons:
- Limited human preference data: The dataset relies on semi-automatic methods for obtaining chosen-rejected pairs, which might introduce some noise or bias.
- Potential for spurious correlations: The format of some tasks, particularly in the reasoning domain, might lead to biases in both human and model judgments.
- Lack of correlation with downstream training: The paper does not establish a clear link between benchmark performance and the effectiveness of RMs in downstream RLHF training.
- Potential for data contamination: There is a risk that some models might have been directly trained on the evaluation datasets, which could affect the validity of the leaderboard rankings.